# Preparation of a New Iron-Carbon-Loaded Constructed Wetland Substrate and Enhanced Phosphorus Removal Performance

**DOI:** 10.3390/ma13214739

**Published:** 2020-10-23

**Authors:** Jie Zhao, Jingqing Gao, Junzhao Liu

**Affiliations:** College of Ecology and Environment, Zhengzhou University, Zhengzhou 450001, China; zhaojie8654@163.com (J.Z.); zzuliujunzhao@163.com (J.L.)

**Keywords:** iron-carbon micro-electrolysis, phosphorus removal, chemical precipitation, environmental economic benefits

## Abstract

Iron-carbon substrates have attracted extensive attention in water treatment due to their excellent processing ability. The traditional iron-carbon substrate suffers from poor removal effects, separation of the cathode and anode, hardening, secondary pollution, etc. In this study, a new type of iron-carbon-loaded substrate (NICLS) was developed to solve the problems of traditional micro-electrolytic substrates. Through experimental research, a preparation method for the NICLS with Fe and C as the core, zeolite as the skeleton, and water-based polyurethane as the binder was proposed. The performance of the NICLS in phosphorus-containing wastewater was analyzed. The results are as follows: The optimal synthesis conditions of the NICLS are 1 g hydroxycellulose, wood activated carbon as the cathode, an activated carbon particle size of 200-60 mesh, and an Fe/C ratio of 1:1. Acidic conditions can promote the degradation of phosphorus by the NICLS. Through the characterization of the NICLS (scanning electron microscope (SEM), X-ray diffractometer (XRD), and energy-dispersive spectrometer (EDS), etc.), it is concluded that the mechanism of the NICLS phosphorus removal is a chemical reaction produced by micro-electrolysis. Using the NICLS to treat phosphorus-containing wastewater has the advantages of high efficiency and durability. Therefore, it can be considered that the NICLS is a promising material to remove phosphorus.

## 1. Introduction

With the increases in population, industrialization, urbanization, and agricultural modernization, water pollution is becoming increasingly serious [1]. A large amount of phosphorus-rich domestic sewage, industrial wastewater, and farmland runoff are discharged into lakes, rivers, oceans, and other water bodies, causing eutrophication of water bodies [2,3,4]. Eutrophication of water bodies poses a huge threat to human health [5]. Therefore, it is very important to remove phosphorus from water.

The commonly used phosphorus removal methods mainly include chemical and biological methods [6]. The chemical precipitation method is widely used in engineering, but this method has a high cost and a large amount of precipitated sludge. Coagulant addition can easily cause equipment corrosion, blockages, and water pollution [7,8]. Biological dephosphorization methods have lower operating costs, but the effectiveness of dephosphorization generally does not exceed 30% [9]. Biological dephosphorization has low stability and strict operation requirements. The effectiveness of dephosphorization is greatly affected by wastewater temperature and pH [9]. Therefore, it is necessary to research and develop new efficient and feasible phosphorus removal technologies. As an alternative to conventional and chemical methods in ecological engineering, the employment of constructed wetlands (CW) to treat phosphorous wastewater is becoming increasingly popular [10]. The main mechanisms for removing phosphorus from water in constructed wetlands include substrate removal, plant removal, and microbial assimilation. A large number of studies have shown that the substrate in constructed wetlands is the main determinant of phosphorus removal [11,12]. The substrate in the constructed wetland provides growth media for plants and microorganisms, and at the same time, can directly remove phosphorus through ion exchange, specific or nonspecific adsorption, complexation, and precipitation. The most important thing is that the constructed wetland substrate contains abundant metal elements, such as iron, aluminum, and calcium. The ions of these metal elements can combine with the phosphate ions in the water to form a precipitate to achieve the purpose of removing phosphorus. Therefore, choosing the appropriate substrate is the key to improving the phosphorus removal capacity of constructed wetlands.

In recent years, iron-carbon (Fe-C) micro-electrolysis technology has been used in many applications for the treatment of phosphorus-containing wastewater [13]. This method is also called the internal electrolysis method and iron-carbon method [14]. Iron-carbon micro-electrolysis technology uses wastewater as the electrolyte, iron in the micro-electrolytic substrate as the anode, and activated carbon as the cathode to form a “primary cell.” The discharge is used to form an electric current to electrolytically oxidize and reduce the wastewater discharge. The principle is based on using the combined effects of electrochemistry, oxidation-reduction, and flocculation to achieve the purpose of removing organic pollutants [15]. Iron-carbon micro-electrolysis has the advantages of a wide application range, low cost of processing, short processing time, convenient operation and maintenance, low power consumption, etc. [16,17].

The preparation technology of iron-carbon (Fe-C) substrates has also evolved greatly. In previous studies, iron-carbon micro-electrolytic substrates were mainly synthesized by physically mixing or sintering iron and carbon as raw materials [18]. In 2014, Zhou et al. used active iron-carbon micro-electrolysis systems to remove organic phosphates from discharged circulating cooling waters [19]; Shi et al. prepared an iron oxide/activated carbon composite adsorbent (activated carbon loaded with iron oxide) that can effectively treat phosphate-contaminated water [20]; Chen et al. synthesized an iron-carbon (Fe-C) micro-electrolysis material substrate through a carbothermal reduction process using flotation waste copper slag as a carbon source and anthracite as a carbon source [21]. Although these substrates have a good removal effect, they have the following problems: (1) The iron scrap landfill treatment time is long, and packing compaction is prone to occur [22]; (2) iron-carbon stacks act as iron-carbon primary cells to obtain a good phosphorus removal effect, but with increasing processing time, cathode and anode separation and blocking problems will occur [23]; and (3) through firing and the formation of a regular structure, a substrate can be obtained with strengthened contact between the cathode and anode to avoid separation problems, but the energy consumption is high, and air pollution is caused during firing [24]. To solve the problems related to the application of traditional iron-carbon micro-electrolysis, improvements must be made. There are two ways to solve the problems of traditional iron-carbon substrates: (1) Combine new technologies, such as oxidation, on the basis of traditional iron-carbon substrates; (2) develop new iron-carbon substrates. However, combining micro-electrolysis with other technologies and developing new reactors will increase the costs of system operation and maintenance to a certain extent and complicate the operation. In this paper, a new type of iron-carbon substrate is prepared, which not only solves the problems of iron powder and carbon powder loss, and cathode and anode separation, but also solves the serious problem of substrate compaction, saves energy, and avoids air pollution.

In this study, a new type of iron-carbon-loaded substrate (NICLS) was prepared for the first time. Using water-based polyurethane as the binder, hydroxyethyl cellulose as the thickener, iron and carbon as the coating, and zeolite as the aggregate, single-factor experiments were used to optimize the NICLS to determine the optimal ratio of the substrate raw materials. A static test was used to study the effects of different factors (pH, initial concentration, reaction time, and temperature) on phosphorus removal by the NICLS. A specific surface area analyzer (BET), scanning electron microscope (SEM), X-ray diffractometer (XRD), and energy-dispersive spectrometer (EDS) were used to determine the specific surface area of the NICLS, analyze its composition, examine its surface structure, analyze the type and content of elements in the substrate, and explore its mechanism of phosphorus removal, laying the foundation for the practical application of this substrate. The innovation of this research lies in the fact that the NICLS was first made by loading iron carbon on zeolite, which not only achieves zeolite recycling but also improves the removal of phosphorus, promotes the development of phosphorus removal substrate materials, improves water quality, and has good environmental and social benefits.

## 2. Materials and Methods

### 2.1. Materials

Zeolite particles, iron powder, and activated carbon (coal activated carbon, wood activated carbon, tar activated carbon, and coconut shell activated carbon) were all obtained from the Zhengzhou Building Materials Market (Zhengzhou, China). The particle size of the zeolite was 10–16 mm, and the surface was smooth. The zeolite was cleaned and dried before use.

### 2.2. Chemicals

All the chemicals used in this study were of analytical reagent grade. Phosphorous solutions with different concentrations were prepared from KH_2_PO_4_, and the pH value of the solutions was adjusted with 0.1 mol/L hydrochloric acid and 0.1 mol/L sodium hydroxide [25]. Water-based polyurethane, hydroxyethyl cellulose, potassium dihydrogen phosphate (KH_2_PO_4_), ammonium molybdate [(NH_4_)_6_Mo_7_O_24_·4H_2_O], concentrated sulfuric acid, ascorbic acid (C_6_H_8_O_6_), potassium persulfate, and potassium antimony tartrate (C_4_H_4_KO_7_Sb·1/2H_2_O) [26] were obtained from Zhengzhou Heshun Technology Co., Ltd. (Zhengzhou, China), and all solutions were prepared using deionized water (the conductivity of deionized water is 1.0 µs/cm).

### 2.3. Preparation of NICLS

The NICLS is composed of basic ordinary zeolite and a cladding layer, which is composed of components (iron and carbon) capable of removing a large amount of phosphorus. The iron-carbon, ordinary zeolite, and binder are mixed to adhere the iron carbon to the ordinary zeolite.

The specific procedure was as follows: First, the zeolite was washed with water. Then, 3.0 g of hydroxyethyl cellulose was added (to increase the viscosity of the water-based polyurethane) to 30 mL of water-based polyurethane, and the mixture was stirred evenly. Then, 20 g of clean zeolite was added, and the mixture was stirred continuously to coat the zeolite with a layer of binder. This mixture was then added to 10 g of iron-carbon polymer, stirred well, and shaken to wrap the iron-carbon on the zeolite. Finally, the material was dried naturally to obtain the NICLS, as shown in Figure 1.

### 2.4. Optimization of NICLS Preparation Conditions

To optimize the NICLS preparation conditions, the removal capacity of phosphorus was taken as the detection target, and a single-factor method was used to study different iron-carbon mass ratios (Fe/C = 4:1, 3:1, 2:1, 1:1, 1:2, 1:3, and 1:4), thickener contents (0.01, 0.1, 0.5, 1.0, and 1.5 g), carbon types (coal, wood, tar, and coconut shell), and carbon particle sizes (>60 mesh, 200-60 mesh, and <200 mesh). The influence of these four factors on the phosphorus removal performance of the NICLS was experimentally determined to obtain the best preparation conditions for the NICLS, and the significant difference analysis of the experimental data.

### 2.5. Comparison with Other Constructed Wetland Substrates

In this experiment, five types of artificial wetland substrates were selected for comparison, namely ceramsite, gravel, volcanic rock, steel slag, quartz sand, etc., all of which come from the building materials market.

Ceramsite is made of natural clay minerals or solid waste as the main raw material, supplemented by a small amount of additives, and mixed and granulated by a sintering process. The ceramsite has the characteristics of large specific surface area and developed porosity, which can realize the removal of phosphate ions in water [27].

Gravel is a natural pellet made of exposed rocks through weathering or long-term transport by water. Gravel has a wide range of sources and low cost. It is one of the most widely used substrates in experiments and engineering at home and abroad [28].

The main components of volcanic rock are SiO_2_, Al_2_O_3_, Fe_2_O_3_, and CaO, and it has good surface activity and pore structure [29].

Steel slag is mainly composed of calcium, iron, silicon, magnesium, and a small amount of oxides such as aluminum, manganese, and phosphorus. It has a pore structure and a large specific surface area. Steel slag has stable performance, low price, and good adsorption capacity, which is used to treat wastewater [30].

Quartz sand is one of the most widely used and most used water purification materials. Quartz sand has stable chemical properties, strong dirt interception ability, and has the advantages of long service life, wide application range, and low processing cost [31].

### 2.6. NICLS Removal Experiment of Phosphorus in Simulated Wastewater

The NICLS was prepared under the optimal conditions (the optimal synthesis conditions of the NICLS are 1 g hydroxycellulose, wood activated carbon as the cathode, an activated carbon particle size of 200-60 mesh, and an Fe/C ratio of 1:1), and experimentally prepared simulated phosphorus solution (0.2197 g KH_2_PO_4_ dissolved in 100 mL water) was used as the treatment object. The influence of different pH values, initial phosphorus concentrations, and reaction times on the treatment effect of the NICLS on wastewater was investigated using a single-factor analysis method, and blank and parallel experiments were carried out throughout the whole process.

The specific procedure was as follows: A total of 0.5 g of the NICLS was weighed into a 250 mL Erlenmeyer flask, 100 mL of a set concentration of phosphorus solution was added, and the pH of the solution was adjusted to 6 with 0.1 mol/L hydrochloric acid or 0.1 mol/L sodium hydroxide solution. The solution was shaken in a constant-temperature shaker at 150 r·min^−1^ and 25 °C for 24 h. A syringe was used to take out 5 mL of the supernatant from the Erlenmeyer flask and filter it with a 0.45 µm filter membrane; the phosphorus concentration in the filtrate was determined by ammonium molybdate spectrophotometry (Multi-parameter water quality analyzer (5B-3B(V8)) of Lianhua Technology, Zhengzhou, China), and the removal capacity of phosphorus on the NICLS was calculated according to Equation (1) [32].
(1)$$A_{e}=\frac{(C0-Ce)V}{m}$$
where *C*_0_ is the initial phosphorus concentration of the solution, mg/L; *A_e_* is the removal capacity at equilibrium, mg/g; *C_e_* is the phosphorus concentration at reaction balance, mg/L; *m* is the mass of the NICLS, g; and *V* is the volume of the phosphorus solution, L.

#### 2.6.1. Effect of Reaction Time and Initial Phosphorus Concentration

Five series of 250 mL Erlenmeyer flasks were prepared: 0.5 g of the NICLS and the phosphorus solution was added to reach the final phosphorus concentrations of 10, 30, 50, 100, or 150 mg/L, and the pH of the solution was adjusted to 6 with 0.1 mol/L hydrochloric acid or 0.1 mol/L sodium hydroxide solution [33]. The flasks were incubated at 25 °C and 150 r·min^−1^, and samples were collected at different times (1, 3, 5, 7, 9, 11, 13, 15, 17, 19, 21, 23, and 24 h). After the water sample was left and centrifuged, the phosphorus content in the solutions was determined by ammonium molybdate spectrophotometry, and the corresponding removal capacity was calculated [34]. The calculation formula used to determine the removal capacity is formula (1).

#### 2.6.2. Effect of pH

Using the same test method of phosphorus removal by the NICLS, we set different pH values (3, 5, 7, 9, and 11) to determine the effect of pH on the NICLS removal of phosphate ions [35]. The calculation formula used to determine the removal capacity is formula (1).

### 2.7. Characterization of the NICLS

A surface area analyzer was used to measure the BET surface area (SBET) and pore volume of the NICLS from the 77 K nitrogen adsorption/desorption isotherm [36]. The pore size was derived using the BET and DFT methods [37]. Scanning electron microscopy (SEM) was used to characterize the surface morphology of the NICLS before and after the reaction, and energy-dispersive X-ray spectroscopy (EDS) was used to determine the composition change in the NICLS before and after the reaction [38,39]. X-ray diffraction (XRD) patterns of the NICLS were recorded with an X-ray diffractometer (China DX-2700), (Cu Kα, 30 mA, 40 kV) [40].

## 3. Results and Discussion

### 3.1. Optimization of NICLS Preparation Conditions

The proportion of the raw materials is the most important part in the preparation of the NICLS and determines the effect of the NICLS on the treatment of simulated wastewater. The capacity of phosphorus removed is an indicator of whether the NICLS has been successfully prepared.

#### 3.1.1. Effect of Adding Varying Amounts of Thickener (Hydroxyethyl Cellulose) on the Removal Capacity of NICLS

It can be seen from Figure 2 that when the addition amount of thickener is 0.01 g, the removal capacity first increases and then decreases sharply. This is because when the amount of thickener is too small, the binding effect of the binder on the iron-carbon is reduced. In the early stage, the iron-carbon coating can fully react with phosphorus in the water, and the amount of iron-carbon bound in the later stage is limited, so the removal amount declines. When the added amount of thickener is greater than 0.01 g, the removal capacity also increases continuously. When the amount of thickener added is 1.0 g, the phosphorus removal capacity reaches the maximum, but when the amount of thickener added is 1.5 g, the removal capacity decreases instead. This is because when there is too much thickener, the binder coating on the zeolite is too thick, resulting in a large mass, and the iron-carbon cannot be completely coated on the zeolite, resulting in a decrease in the amount of removal; furthermore, the resulting iron-carbon substrate is not strong enough, is easy to slag, and is difficult to use as a substrate in practical applications. Therefore, in this study, 1.0 g was chosen as the optimal amount of thickener. In order to further prove that adding 1.0 g of hydroxyethyl cellulose is best, we used analysis of variance to explore the significant effect. The results are shown in Table 1.

It can be seen from Table 1 that the difference in the amount of hydroxyethyl cellulose added after 5 h has a significant effect.

#### 3.1.2. Effects of Activated Carbon Types and Activated Carbon Particle Size on the Removal Capacity of NICLS

Iron-carbon micro-electrolysis technology generally uses iron-based materials and carbon-based materials as electrode materials, immersed in an aqueous electrolyte solution, to form a large number of microscopic galvanic cells [41]. Therefore, the parameters (type and particle size) of the activated carbon raw materials are important parameters that can significantly affect the phosphorus removal efficiency in the process.

To study the effect of various types of activated carbon on the phosphorus removal performance, the dosage of the NICLS and the Fe/C ratio were fixed at 5 g L^−1^ and 1:1, respectively. It can be seen from Figure 3a that the removal rate of phosphorus by wood activated carbon is higher than those of the other three activated carbons. The pore size distribution and pore structure of activated carbon depend on the type of activated carbon. In previous studies, wood activated carbon was found to have a large specific surface area and a well-developed porous structure, providing many active sites for phosphorus removal [42]. Therefore, wood activated carbon was the activated carbon selected in this experiment. It can be seen from Figure 3b that activated carbon with a particle size of 200-60 mesh has a better phosphorus removal effect. For a given Fe/C, the particle size of the carbon has a strong influence on the micro-electrolysis reaction. The smaller the particle size of the carbon, the larger the specific surface area of the NICLS formed [36]. The more iron-carbon contact there is in the wastewater, the more primary batteries are formed, and the faster the micro-electrolysis reaction rate. However, if the particle size is too small, the iron-carbon supported on the zeolite will be too thick, which is not conducive to a full reaction. It can be seen that the particle size of the activated carbon should not be as small as possible and that activated carbon with a moderate particle size has a higher phosphorus removal rate. Therefore, the particle size of activated carbon selected from this test was 200-60 mesh.

In order to further prove the optimal conditions of activated carbon, this paper uses analysis of variance to explore the significant impact of activated carbon. The results are shown in Table 2 and Table 3.

It can be seen from Table 2 and Table 3 that the type of activated carbon and the particle size of activated carbon have obvious differences in the removal of phosphorus, so it is further proved that the particle size of wooden activated carbon and activated carbon of 200 mesh-60 mesh is the best phosphorus removal condition.

#### 3.1.3. Effect of Fe/C Ratio of NICLS on Removal Capacity

Iron serves as the anode, and activated carbon serves as the cathode. Different iron/carbon values result in different contact area ratios between the anode and cathode, which significantly affect the corrosion rate and phosphorus removal rate [43]. As shown in Figure 4, when the Fe/C ratio is 1:1, the phosphorus removal effect is best. It can be seen that too large or too small an Fe/C ratio is detrimental to the treatment effect; when the Fe/C ratio is too small, there are fewer iron-carbon macro-primary cells, and too much activated carbon reduces the contact probability between the sewage and electrode reaction active products, decreasing the electrode reaction rate and resulting in a poor treatment effect. In addition, due to removal by activated carbon, too much activated carbon will inhibit the electrode for the primary battery reaction [44]. When the Fe/C ratio is too large, the number of macro primary batteries is insufficient, resulting in a reduction in the treatment effect. As activated carbon is not consumed in the reaction, iron filings are continuously reacted and consumed. To a certain extent, increasing the proportion of iron will increase the removal rate, but excessive iron filings will agglomerate, decreasing the reaction surface area and hence the phosphorus removal rate [45]. Considering these factors comprehensively, the optimal value of the Fe/C ratio under the test conditions was 1:1.

In order to further prove the optimal conditions of Fe/C ratio, this paper uses analysis of variance to explore the significant influence of Fe/C ratio.

It can be seen from Table 4 that different Fe/C ratios have obvious differences in the removal of phosphorus, so it is further proved that an Fe/C ratio of 1:1 is the best phosphorus removal condition.

### 3.2. Comparison of the Phosphorus Removal Effects of NICLS, Coated Iron, Coated Carbon, and Uncoated Zeolite

It can be seen from Figure 5 that the phosphorus removal effect by iron alone is minimal. With prolonged time, the removal of phosphorus increases, which may be because a very small amount of Fe is converted into Fe_2_O_3_ in the presence of oxygen, and Fe_2_O_3_ can adsorb PO_4_^3−^, thereby removing a very small amount of phosphorus [46]. The phosphorus removal effect by activated carbon alone is not significant. The removal amount of phosphorus by this method is slightly higher than that of the iron method. This may be because activated carbon has a certain adsorption capacity and can adsorb a small amount of phosphorus, but the effect is minimal [47]. With the use of iron and activated carbon to form a micro-electrolysis electrode, the removal effect of phosphorus is significant, and the removal of phosphorus reaches 8 mg/g; this result occurs because the micro-electrolysis system produces a large amount of Fe^2+^ and Fe^3+^, and iron ions can react with PO_4_^3−^ to form phosphate precipitates to achieve phosphorus removal [48,49]. In addition, under neutral or alkaline conditions, Fe^3+^ will form Fe(OH)_3_. Fe(OH)_3_ can promote the coagulation and precipitation of colloids, thereby removing phosphorus [50]. It can be seen from these results that the methods using iron alone and activated carbon alone have a phosphorus removal capacity that is much smaller than that of iron-carbon micro-electrolysis. The phosphorus removal by iron-carbon micro-electrolysis is not a simple superposition of the removal rates obtained with iron filings and activated carbon individually. In addition to adsorption on activated carbon, this process also involves the galvanic role of iron-carbon electrodes: The anode of the micro-electrolysis system produces a large amount of Fe^2+^ and Fe^3+^, and iron ions form phosphate, Fe(OH)_3_, etc., which can all play a role in phosphorus removal. Therefore, the removal process of phosphorus is the result of the combined action of physical and chemical effects, among which the primary battery role of the iron-carbon electrode plays a major role. Compared with the loaded substrate, the amount of phosphorus adsorbed by the zeolite is very small because the pore area of the zeolite itself is small and the adsorption effect is poor.

In order to further prove that the NICLS has the best phosphorus removal effect, this paper uses analysis of variance to explore the analysis of the significant influence of Fe, C, NILS, and zeolite on phosphorus removal.

It can be seen from Table 5 that Fe, C, NICLS, and zeolite have significant differences in the removal of phosphorus, so it further proves that the NICLS has the best phosphorus removal effect.

### 3.3. Comparison with Other Constructed Wetland Substrates

The removal effect of seven kinds of substrates on TP (After digestion, various forms of phosphorus are converted into orthophosphate) in simulated wastewater is shown in Figure 6. After 24 h of reaction, the removal capacity of seven kinds of substrates for phosphorus is as follows: NICLS > steel slag > quartz sand > volcanic rock > ceramsite > zeolite > gravel. Among them, the removal capacity of the NICLS, steel slag, quartz sand, volcanic rock, ceramsite, zeolite, and gravel on phosphorus in sewage are 7.78, 4.32, 2.44, 1.08, 0.82, 0.56, and 0.2 mg/g, respectively. The experimental results show that the NICLS has the best removal effect on phosphorus in sewage.

### 3.4. NICLS Phosphorus Degradation Ability

#### 3.4.1. Effect of Contact Time and Initial Phosphorus Concentration

Figure 7 shows the relationship between phosphorus removal by the NICLS and reaction time at different initial phosphorus concentrations of 10, 30, 50, 100, and 150 mg/L. Within the first 5 h, the removal capacity corresponding to different initial phosphorus concentrations shows an upward trend, and then the removal process becomes very slow and eventually reaches equilibrium. This may be because at the beginning of the reaction, iron-carbon contact is more complete, redox and other reactions occur more fully, more Fe^2+^ is generated at the anode, more Fe^3+^ is oxidized, and there are sufficient adsorption sites on the surface of the activated carbon, which is more conducive to the removal of phosphorus [15,51]. However, as the reaction time increases, the surface of the NICLS is passivated, and therefore, the removal of phosphorus no longer increases significantly. As seen from Figure 7, for the 10 mg/L phosphorus solution, the removal of phosphorus by the NICLS takes only 5 h to reach equilibrium; for the 30 mg/L phosphorus solution, it takes 9 h to reach equilibrium; for the 50 mg/L phosphorus solution, equilibrium can be reached within 13 h; for the 100 mg/L phosphorus solution, it takes 18 h to reach equilibrium; while the 150 mg/L phosphorus solution needs 21 h. Obviously, an increase in initial phosphorus concentration leads to a significant increase in equilibrium time and removal capacity [52].

#### 3.4.2. Effect of pH

The pH value is one of the important factors that affect the processing efficiency of the micro-electrolysis system. Figure 8 shows the removal rate of phosphate by the NICLS at different pH values. The removal rate of phosphate ions under acidic conditions is higher than that under alkaline conditions. A higher phosphate removal rate can be obtained under acidic conditions, which is consistent with the results of phosphate treatment with a fired iron-carbon substrate [41]. The removal of TP in simulated wastewater by the NICLS mainly involves adsorption by activated carbon and the electrolytic reaction. The activated carbon in the NICLS has an adsorption effect on phosphorus in simulated wastewater, but according to the test results in the early stage of these experiments, its adsorption effect is quite small. Under acidic conditions, the NICLS anode reaction is more likely to generate a large amount of Fe^2+^ in the solution, and then Fe^2+^ is oxidized to Fe^3+^, which may be removed by further reactions to generate iron phosphate precipitation [53]. Under alkaline conditions, excess OH^–^ in the solution will compete with phosphate ions for Fe^3+^, and a decrease in the concentration of iron ions will occur due to the combination of iron ions and OH^–^ to form Fe(OH)_3_, which can promote the coagulation and precipitation of colloids, thereby removing phosphorus [54]. An excessive pH value enriches the surface of activated carbon with a large amount of negative charge, and electrostatic repulsion makes it difficult for phosphate or hydrogen phosphate ions to access the surface of the activated carbon particles, resulting in a decrease in the removal rate of TP in wastewater by the NICLS; furthermore, alkaline conditions are not conducive to the progress of the electrolysis reaction (the effect of activated carbon is small). Therefore, under alkaline conditions, the NICLS has a poor removal effect on TP in simulated wastewater. In the neutral range, iron is mainly in the form of ferrous ions or iron ions, and it is more likely to combine with phosphate ions to form precipitates [55].

### 3.5. Characterization of NICLS before and after the Reaction

#### 3.5.1. Specific Surface Area of NICLS

Table 6 lists the specific surface areas of uncoated and coated substrates. The data show that coating the iron and carbon helps increase the specific surface area of the substrate. The specific surface area increased from 2.6231 to 3.7107 m^2/^g. The small pore volume is also consistent with the previous removal experiment results, which also confirm that adsorption by the NICLS is very small. These results affirm the beneficial effect of wrapping iron carbon.

#### 3.5.2. SEM Analysis of NICLS

NICLS samples before and after testing were analyzed by SEM, and the results are shown in Figure 9. Figure 9a,c show that before removing phosphorus, the surface of the NICLS is rough and dense. It can also be observed that there are small particles on the surface of the NICLS, which results in a higher specific surface area, and these particles also increase the number of contact points in aqueous solution. As shown in Figure 9b,d, after removing phosphate, iron ions are eluted due to the micro-electrolysis of the iron-carbon primary battery, resulting in an uneven surface of the NICLS, significantly increased voids, and increased internal structure looseness. There are also many attachments on the surface of the NICLS. Such substances are not observed in the SEM photos of NICLS samples before the experiment, which shows that the attachments are generated during the experiment. These attachments may be Fe(OH)_3_ precipitates formed by Fe^3+^ in the micro-electrolysis process or by reaction with phosphate ions. These precipitates may hinder the micro-electrolysis reaction, resulting in a reduced removal efficiency of phosphorus.

#### 3.5.3. EDS Analysis of NICLS

Figure 10 shows EDS graphs of the NICLS before and after the reaction. As shown in Figure 10a, the main element on the surface of the NICLS is Fe, whose content is much higher than those of other elements. After reaction with phosphorus-containing solution, the proportion of atomic iron decreases, and the proportion of phosphorus increases (Figure 10b), indicating that the iron and carbon in the NICLS undergo micro-electrolysis in phosphorus-containing wastewater. The iron in the substrate forms Fe^2+^ and Fe^3+^, which are released into the aqueous solution. Some of this Fe^2+^ and Fe^3+^ reacts with phosphate ions in the aqueous solution to form FePO_4_ precipitates, and some reacts with OH^–^ in water to generate Fe(OH)_3_ precipitates [49]. Some of these deposits are released to the aqueous solution, and some are attached to the surface of the NICLS, so the atomic ratio of phosphorus on the NICLS surface increases.

#### 3.5.4. XRD Analysis of NICLS

To analyze the prepared NICLS compounds before and after the reaction, XRD tests were carried out, and the results are shown in Figure 11. After the phosphorus-containing NICLS reaction, the types of diffraction peaks are mainly for iron-containing compounds, which are mainly 25.8° FePO_4_ diffraction peaks, that is, the product of the reaction between the NICLS and phosphorus in the solution is mainly FePO_4_. This result indicates that the orthophosphate in the water co-precipitates with the iron ions released by NICLS micro-electrolysis, which can also indicate that the deposits on the surface of the NICLS after the reaction are mainly precipitates formed by phosphate and iron ions.

### 3.6. Mechanism Analysis of NICLS Phosphorus Removal

Based on the previous test results and relevant literature, combined with various characterization methods (SEM, XRD, and EDS of the NICLS before and after the reaction), phosphorus may be removed by precipitation, adsorption, and co-precipitation. The related mechanism is briefly illustrated in Figure 11, and the related chemical reactions are (2)–(6) [49,56].

The iron-carbon micro-electrolysis method, also known as the iron-carbon method, iron reduction method, zero-valent iron method, internal electrolysis method, etc., uses low-potential iron as the anode and high-potential inert carbon (activated carbon, coke, or graphite) as the cathode, while treated sewage is used as an electrolyte solution; in this method, the sewage is treated by countless tiny primary batteries in the solution using the metal corrosion principle [57]. The removal of phosphorus by the NICLS is based on iron-carbon micro-electrolysis. Fe loses two electrons to form Fe^2+^, and Fe^2+^ gains one electron and converts it to Fe^3+^, which can combine with PO_4_^3–^ to achieve phosphorus removal. The principle is as follows:

Anode:(2)$${\mathrm{Fe}\;-\;2e}^{-}\;\to {\;\mathrm{Fe}}^{2+\;}\;\;\;\;E^{0}\;\left({\mathrm{Fe}}^{2+}/\mathrm{Fe}\right)\;=\;-0.44\;v$$
(3)$${\mathrm{Fe}}^{2+}\;+\;e^{-}\;\to {\;\mathrm{Fe}}^{3+}\;\;\;\;E^{0}\;\left({\mathrm{Fe}}^{3+}\;/{\mathrm{Fe}}^{2+}\right)\;=\;-0.77\;v$$

Cathode:(4)$${2H}^{+}{\;+\;2e}^{-}\to \;2\;[H]\;\to {\;H}_{2}\;\;\;\;E^{0}\;\left(H^{+}/H_{2}\right)\;=\;0.00\;v$$
(5)$$2\mathrm{Fe}\;+\;6H^{+}\to \;3H_{2}\;+\;2{\mathrm{Fe}}^{3+}\;\;\;\;E^{0}\;\left({\mathrm{Fe}}^{3+}/\mathrm{Fe}\right)\;=\;-1.21\;v$$

The phosphorus co-precipitation reaction formula is as follows:(6)$${\mathrm{Fe}}^{3+}\;+\;{{\mathrm{PO}}_{4}}^{3-}+\;2H_{2}O\;\to \;{\mathrm{FePO}}_{4}.2H_{2}O\;↓$$

In addition to the above chemical precipitation process, the following processes occur (Figure 12). Iron coagulation: The anode produces a large amount of Fe^2+^ and Fe^3+^ during the micro-electrolysis process, which are very good flocculants and will be converted into Fe(OH)_2_ under appropriate pH conditions. If oxygen is present, the iron will also be converted into Fe(OH)_3_, and a mixture of nFe(OH)_2_ and mFe(OH)_3_ will also be formed; the resulting flocs can neutralize the colloidal charge; promote colloidal collision, coagulation, and precipitation; and promote the removal of phosphorus (this may also be the phosphorus removal mechanism under alkaline conditions). Physical adsorption: The activated carbon of the cathode is also a good adsorbent with a certain adsorption capacity for phosphorus (this was also proven in the preliminary test) [57].

### 3.7. Benefit Analysis

Zeolite is naturally pollution-free, has a wide range of resources, has a large amount of reserves, and is cheap and easy to obtain. However, zeolite has poor phosphorus removal performance [58]. Putting zeolite directly in a constructed wetland as a substrate cannot achieve good results for phosphorus removal.

At present, the traditional iron-carbon micro-electrolytic substrate is prone to electrode separation, unstable operation efficiency, and high operation cost and will cause environmental pollution and other problems. Using the NICLS can not only reduce environmental pollution and make full use of zeolite but also save energy and reduce greenhouse gas emissions. This method meets the goals of resource utilization and harmlessness; achieves the perfect combination of environmental benefits, economic benefits, and social benefits; and can be used to embark on a new path of sustainable development. Therefore, in practical applications, the NICLS prepared here can be used as a substrate for artificial wetlands in combination with other materials.

## 4. Conclusions

In this study, a NICLS was successfully synthesized using iron powder, carbon powder, and binder. The prepared NICLS not only has a strong ability to remove phosphorus in water but also solves the problems in the phosphorus removal process of traditional iron-carbon substrates and provides theoretical guidance for the preparation of new substrates. The research results obtained are as follows: (1) The optimal synthesis conditions of the NICLS are as follows: Hydroxycellulose content of 1 g, wood activated carbon as the cathode, activated carbon particle size of 200-60 mesh, and Fe/C ratio of 1:1. (2) The specific surface area of the NICLS is 3.7107 m^2^/g, and the pore volume is 0.006037 cm^3^/g. (3) Acidic conditions are beneficial for the removal of phosphorus by the NICLS. (4) The phosphorus removal mechanism of the NICLS mainly involves micro-electrolysis. In this micro-electrolysis system, Fe can be oxidized to Fe^3+^, while PO_4_^3–^ can chemically precipitate with Fe^3+^. (5) Comparing the individual removal effects of iron, carbon, and the NICLS on phosphorus, the removal effect of the NICLS on phosphorus is not the simple addition of the effects of iron and carbon, but the result of the combined action of physical and chemical effects.

## Figures and Tables

**Figure 1 materials-13-04739-f001:**
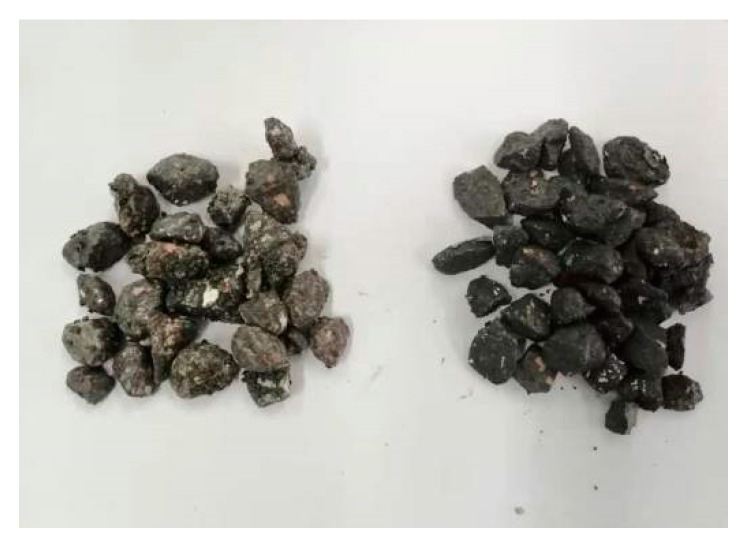
New iron-carbon-loaded substrate.

**Figure 2 materials-13-04739-f002:**
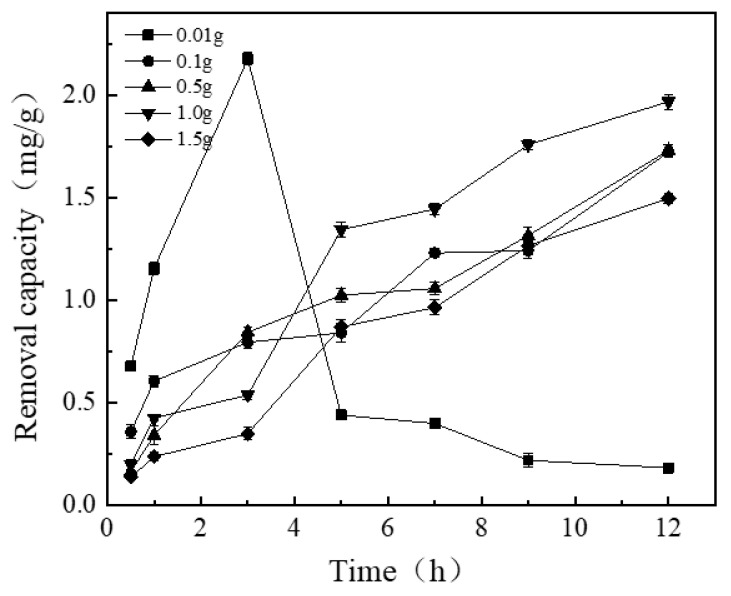
Effect of adding amount of thickener on removal capacity of new type of iron-carbon-loaded substrate (NICLS).

**Figure 3 materials-13-04739-f003:**
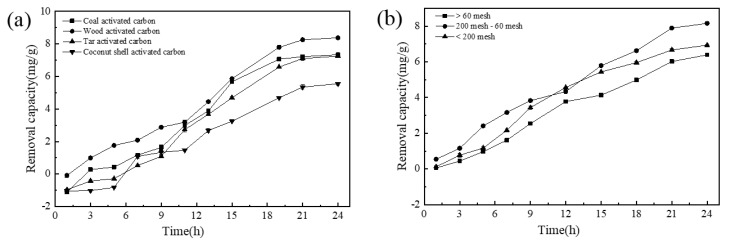
(**a**) Effect of activated carbon type on NICLS removal capacity; (**b**) effect of activated carbon particle size on NICLS removal capacity (activated carbon is wood activated carbon).

**Figure 4 materials-13-04739-f004:**
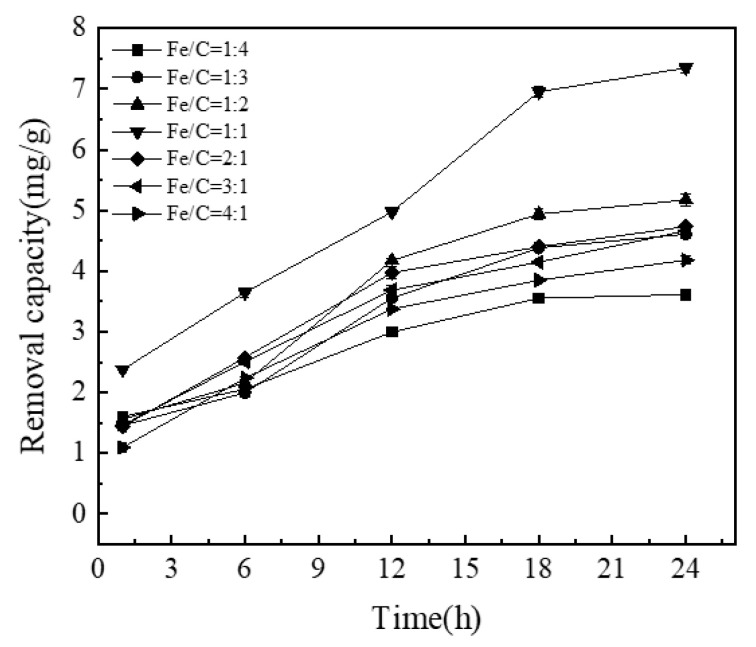
Effect of Fe/C ratio on adsorption capacity of NICLS (activated carbon is wood activated carbon).

**Figure 5 materials-13-04739-f005:**
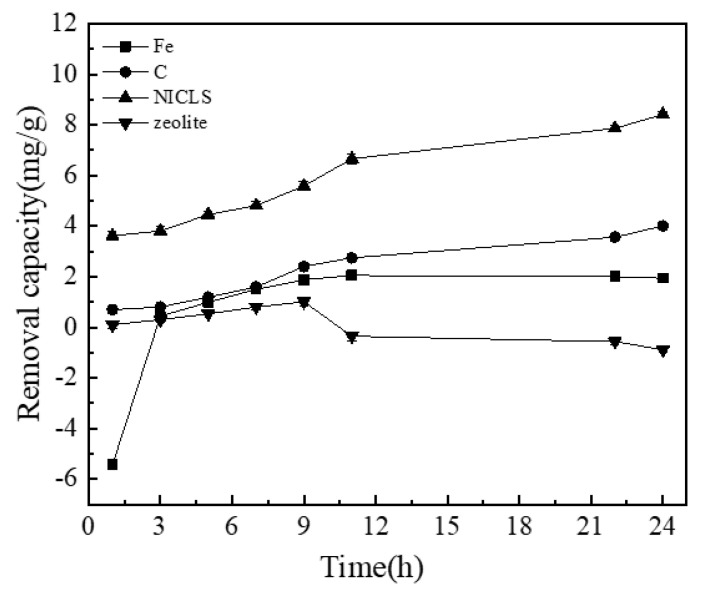
Comparison of the phosphorus removal effects of NICLS, coated iron, coated carbon, and uncoated zeolite (activated carbon is wood activated carbon).

**Figure 6 materials-13-04739-f006:**
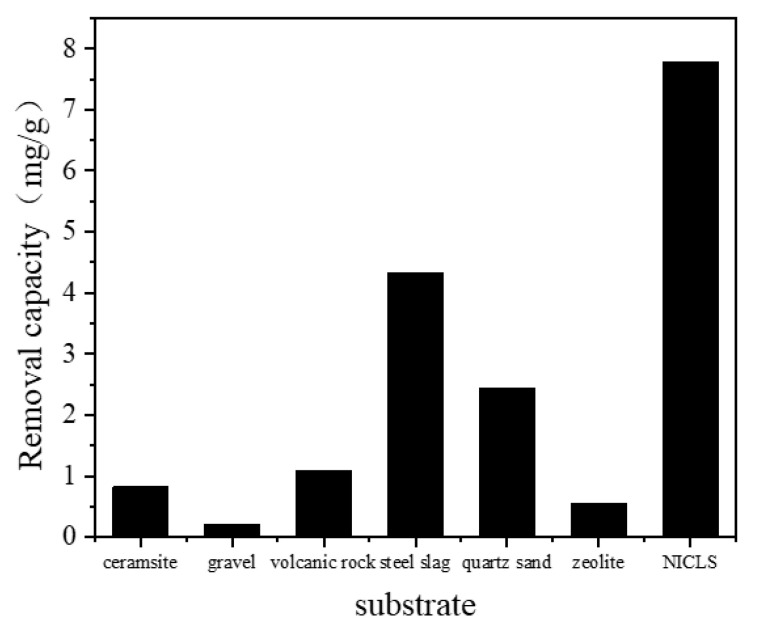
Comparison of phosphorus removal performance of seven types of constructed wetland substrates.

**Figure 7 materials-13-04739-f007:**
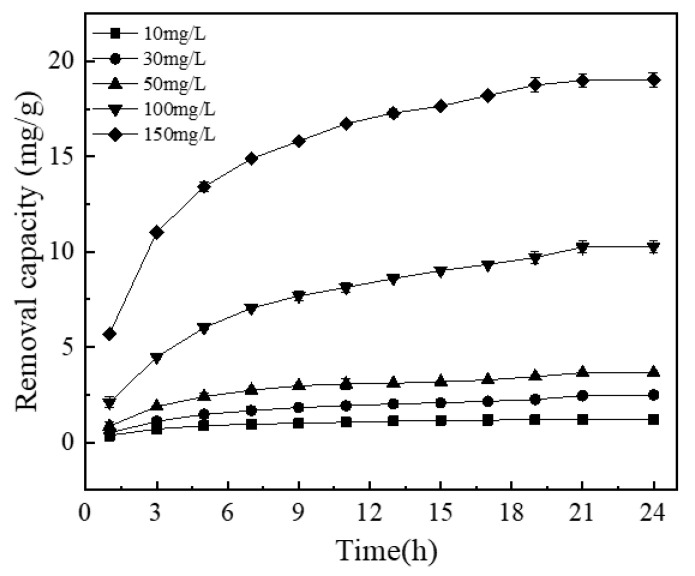
Influence of initial concentration and reaction time on TP removal effect (the dosage is 5 g/L, ph = 7).

**Figure 8 materials-13-04739-f008:**
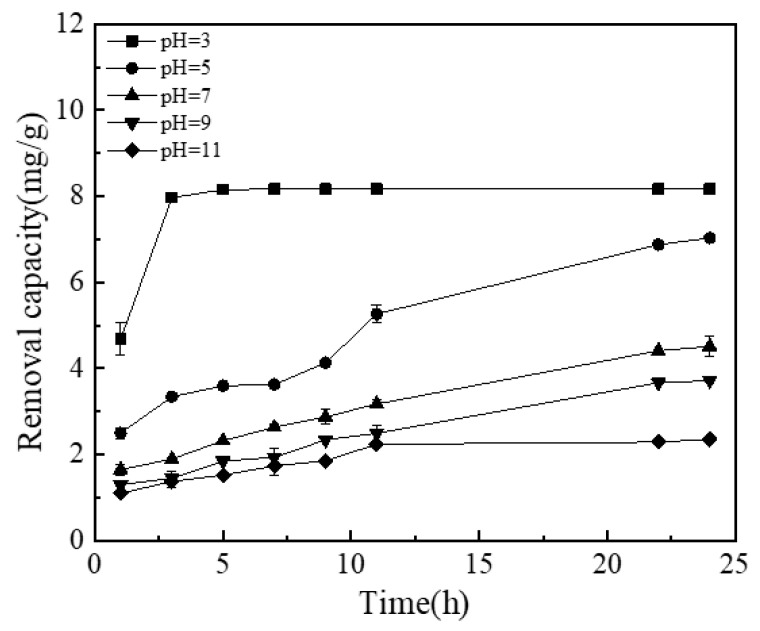
Effect of pH on TP removal effect (the dosage is 5 g/L; initial phosphorus concentration is 50 mg/L).

**Figure 9 materials-13-04739-f009:**
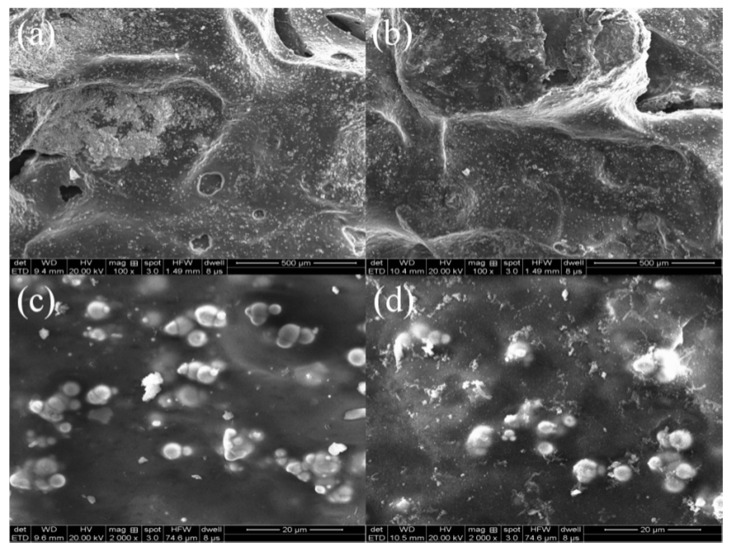
SEM images of the NICLS surface before (**a**,**c**) and after (**b**,**d**) the reaction.

**Figure 10 materials-13-04739-f010:**
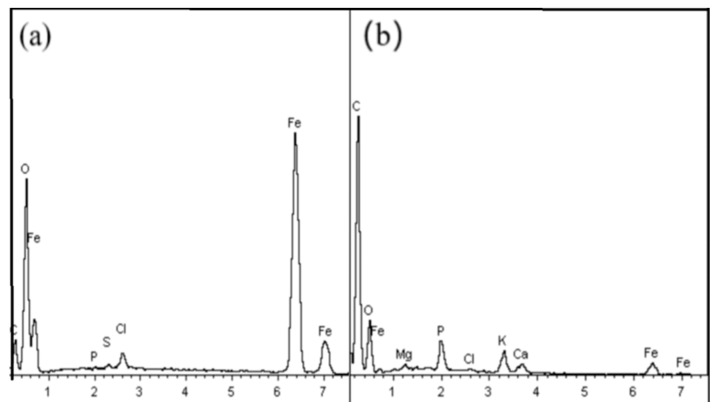
EDS diagram of NICLS surface before (**a**) and after (**b**) reaction.

**Figure 11 materials-13-04739-f011:**
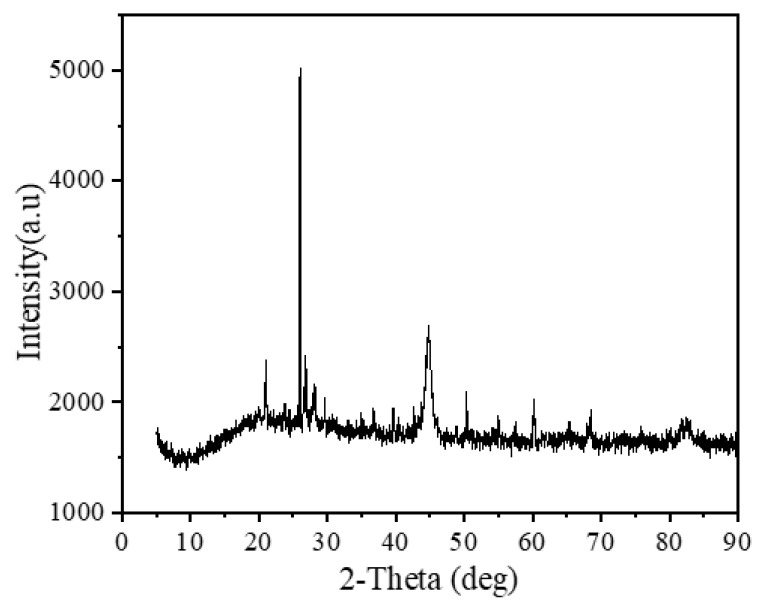
XRD analysis of the reaction between NICLS and phosphorus.

**Figure 12 materials-13-04739-f012:**
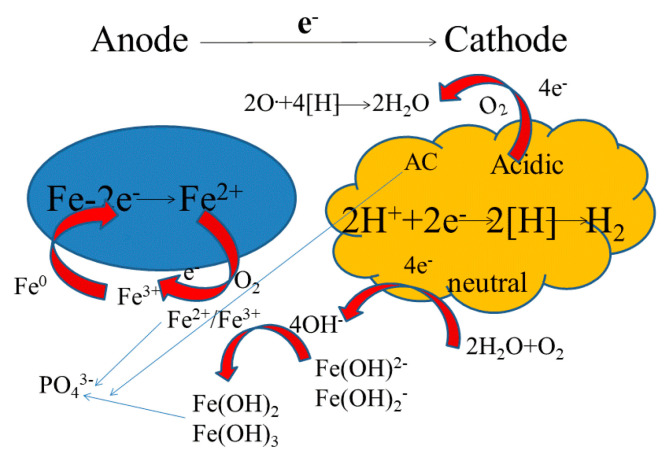
Mechanism of phosphorus removal by NICLS.

**Table 1 materials-13-04739-t001:** Significance analysis of variance.

Time	0.01	0.10	0.50	1.00	1.50
5 h	A	B	C	D	B
7 h	A	B	C	D	E
9 h	A	B	B	C	B
11 h	A	C	C	D	B

(Different letters (A, B, C, D, E) have significant effects).

**Table 2 materials-13-04739-t002:** Variance significance analysis of activated carbon types.

Time	Coal Activated Carbon	Wood Activated Carbon	Tar Activated Carbon	Coconut Shell Activated Carbon
1 h	A	B	C	C
3 h	C	D	B	A
5 h	C	D	B	A
9 h	C	D	A	B
11 h	C	D	B	A
15 h	C	D	B	A
21 h	B	D	B	A
24 h	B	D	B	A

(Different letters (A, B, C, D) have significant effects).

**Table 3 materials-13-04739-t003:** Significance analysis of variance of activated carbon particle size.

Time	>60 Mesh	200 Mesh-60 Mesh	<200 Mesh
1 h	A	B	A
3 h	A	C	B
5 h	A	C	B
7 h	A	C	B
9 h	A	C	B
12 h	A	B	C
15 h	A	C	B
18 h	A	C	B
21 h	A	C	B
24 h	A	C	B

(Different letters (A, B, C) have significant effects).

**Table 4 materials-13-04739-t004:** Significance analysis of variance of Fe/C ratio.

Time	Fe/C = 1:4	Fe/C = 1:3	Fe/C = 1:2	Fe/C = 1:1	Fe/C = 2:1	Fe/C = 3:1	Fe/C = 4:1
1 h	B	B	B	C	B	B	A
6 h	A	A	B	D	C	C	B
12 h	A	B	D	E	D	C	B
18 h	A	C	D	E	C	C	B
24 h	A	A	B	C	B	A	A

(Different letters (A, B, C, D, E) have significant effects).

**Table 5 materials-13-04739-t005:** Variance significance analysis of Fe, C, NICLS, and zeolite phosphorus removal.

Time	Fe	C	NICLS	Zeolite
1 h	A	C	D	B
3 h	A	B	C	A
5 h	B	B	C	A
7 h	B	B	C	A
9 h	B	C	D	A
11 h	B	C	D	A
22 h	B	C	D	A
24 h	B	C	D	A

(Different letters (A, B, C, D) have significant effects).

**Table 6 materials-13-04739-t006:** NICLS property parameters.

Specific Surface Area (before)	Specific Surface Area (after)	Pore Volume	Aperture
2.6231 m^2^/g	3.7107 m^2^/g	0.6037 cm^3^/g	8.4845 nm

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
