# Peer review of "Preparation of a New Iron-Carbon-Loaded Constructed Wetland Substrate and Enhanced Phosphorus Removal Performance"

_materials, 2020, doi:10.3390/ma13214739_

Round 1

Reviewer 1 Report

The article is interesting and presents some actual scientific results. Some coments refers to:

  • 2.1. More information’s about zeolite particles used. 
  • 2.2. not 2.1...use some references here.
  • 2.5. more information’s about substrates used for comparison. 
  • 2.6. some information’s about the used devices for analysis parameters.
  • 2.6.1. and 2.6.2. use some references.
  • At figures 2,3,5,7, and 8 please increase de dimensions and/or tried to evidence the graphics lines difference. 

Reviewer 2 Report

I have examined the manuscript entitled “Preparation of a New Iron-Carbon Loaded 2 Constructed Wetland Substrate and Enhanced 3 Phosphorus Removal Performance” dealing with nutrient removal using modified C-Fe substrate. The paper is interesting as it uses easily available and cheap materials for coagulation/adsorption of phosphorus but some of the issues listed below require attention prior further processing of the paper:

  • Introduction is quite focused on micro-electrolysis however in the entire paper no electrolysis apparatus is not described. Coagulation and adsorption would be more appropriate
  • Conductivity of deionized water must be included
  • Description in section 2.3. is very vague and does not allow to reproduce the experiment. More precise descriptions like weight load of materials, volumes of liquids, volumes of water etc. must be included
  • On what basis was initial P concentration selected? This must be included
  • Authors use Fe in their substrate and refer to various pH behavior. It must be supported and discussed with Pour-Baix diagram of Fe.
  • Section 2.5 describes use of five substrates but section 3.3 seven substrates. What is the reason of this inconsistence
  • Line 301 – what does it mean?
  • Chapter numbering is shoddy
  • Despite the fact that paper talks about micro-electrolysis, adsorption adsorption and coagulation are the main processes the manuscript deals with. Adsorption properties of the prepared material is missing (data in table 1 are insufficient). Langmuir-Hinshelwood model must be included and properly discussed – that’s the absolute base of such kinds of experiments
  • Format of chemical equations and literature is different from the rest of the paper.
  • Description of reactions supports authors’ micro-electrolysis hypothesis, however, no external source of potential is neither described nor used – this means that no electrolysis (even in microscale) occurs. The principle is rather coagulation with this respect the entire paper (introduction, experimental, results and discussions, conclusions) must be rewritten.

With respect to observations summarized above the paper must be revised prior possible publication.

Reviewer 3 Report

In this paper, authors developed a new iron-carbon loaded-substrate (NICLS) by loading iron carbon on zeolite, to overcome the problems of traditional micro-electrolytic substrates for improving phosphorus removal from wastewater and reducing environmental impact. They evaluated the performance of NICLS in phosphorus-containing wastewater to find optimal synthesis and operating conditions.

Broad comments

The paper is very interesting, well-structured and easy to read. Comprehensive state of the art is provided in the introduction and research aim is clearly defined. The material and methods section is impressive: the approach used to investigate the topic is exhaustive. Following, the results and discussion sections provide the relevant outputs of the research and highlight the implications in wastewater treatment to remove excess phosphorus minimizing additional environmental pollution.

I have one main concern regarding the results analysis. As far as is explained in the methodology section, the authors performed parallel experiments at the varying conditions tested which, if I correctly understood and as evident from the figures, means that they have replicates for the different values of tested parameters. If so, I think that the paper would benefit from a statistical analysis of the results for example by using ANOVA to evaluate if differences found are of statistical significance. This would help explain the results shown in the graphs. One example is Figure 2 where the concentration of 1.0 g appears to be the most suitable for an effective treatment but is this difference from the others truly significant? I think the statistical analysis will say yes because standard deviation seems very small, but you need a test that enables to say yes or not. I suggest use ANOVA.

Second, in Figures 3b and 4 the graph related to coconut shell is provided. Why not report the results for wood activated carbon that was the best substrate? Otherwise it is misleading the reader. Similarly specify in the legend of Figure 5 if NICLS was made with wood activated carbon. Anyway if not, I would suggest to report the graph for wood activated carbon.

Several minor English or typing mistakes are present. I would recommend proof reading the manuscript paying attention to them.

Specific comments

Line 39: ecological environment. Remove ecological since it’s useless.

Lines 46-47: for which process? Biological or chemical? Please specify

Line 119: etc? To what refers? In this case I would put the full list

Line 126-127:… to a certain amount…What amount? Please specify

Line 137: check the ratios

Line 146: NCILS was prepared under optimal conditions. Please specify the conditions

Line 153: adjust the pH…to what pH?

Line 155: certain amount..what amount?

Figure 2: the scale of the horizontal axis should start from zero

Round 2

Reviewer 2 Report

I have examined the revisions made by the authors and my opinion is that they put significant effort to the improvement of the manuscript. All of my comments were either satisfactory addressed or justified. My recommendation is to accept the paper in present form.

Reviewer 3 Report

Dear authors

the manuscript has been greatly improved and the quality at present is very high.

Regards.